# The Role of COVID-19-Associated Fear, Stress and Level of Social Support in Development of Suicidality in Patients Diagnosed with Affective and Stress-Induced Psychiatric Disorders during the COVID-19 Pandemic—A Comparative Analysis

**DOI:** 10.3390/brainsci13050812

**Published:** 2023-05-17

**Authors:** Dusan Kuljancic, Mina Cvjetkovic Bosnjak, Djendji Siladji, Darko Hinic, Dunja Veskovic, Nebojsa Janjic, Dragana Ratkovic, Olga Zivanovic, Vesna Vasic, Branislav Sakic

**Affiliations:** 1Medical Faculty, University of Novi Sad, Hajduk Vejkova 3, 21000 Novi Sad, Serbia; 2Clinic of Psychiatry, University Clinical Centre of Vojvodina, Hajduk Veljkova 4, 21000 Novi Sad, Serbia; 3Faculty of Science, Department of Psychology, University of Kragujevac, Radoja Domanovića 12, 34000 Kragujevac, Serbia; 4Clinic of Dermatology, University Clinical Centre of Vojvodina, Hajduk Veljkova 4, 21000 Novi Sad, Serbia; 5Clinic of Gastroenterology and Hepatology, University Clinical Centre of Vojvodina, Hajduk Veljkova 4, 21000 Novi Sad, Serbia

**Keywords:** COVID-19, fear, anxiety disorders, social support, affective disorders, suicide, resilience, public health crisis, social isolation, mental health

## Abstract

Only a few studies seem to address suicidality as an effect of the COVID-19 pandemic in persons previously affected by psychiatric disorders. The relationship between fear and stress caused by the COVID-19 pandemic and the level of social support and suicidality in patients diagnosed with affective and stress-induced psychiatric disorders prior to the onset of the COVID-19 pandemic were investigated. This study was observational and involved 100 participants. The examined period was from April 2020 to April 2022. The Fear of COVID-19 Scale (FCV-19S), the Oslo Social Support Scale 3 (OSSS-3) and general psychiatric interviews were used to obtain data. A statistically significant relationship between the impact of COVID-19-related distress on the occurrence of suicidality and the year of the pandemic χ^2^(2, N = 100) = 8.347, *p* = 0.015 was observed. No statistically significant correlation was found between suicidal behavior, stress intensity, fear and the score on the social support scale (*p* > 0.05). Fear related to the COVID-19 pandemic can only be seen as a contributor to suicidality. Overall, social support does not always act protectively. Previously stressful experiences such as wars, poverty and natural disasters seem to play a significant role in the resilience to each new public health crisis.

## 1. Introduction

The COVID-19 pandemic represents a unique event in modern world history. In addition to the fact that COVID-19 is a threat to physical health, it has been shown that the pandemic is a general threat to mental health and the overall quality of life of each individual [1]. The COVID-19 pandemic itself, as well as all the epidemiological measures introduced to contain it, represent a psychological burden for the population, disrupting the personal, familial and social functioning of the individual, especially those in vulnerable social groups, such as psychiatric patients, who are often on the margins of society even under the usual circumstances [2,3].

Severe epidemiological restrictions implemented to control the spread of the SARS-CoV-2 virus caused a drastic disruption of established life patterns, restriction of freedom of movement, social isolation, loss of intimate social contacts, fear of infection, death, feelings of tension and anxiety caused by uncertainty and insecurity in every aspect of human life [2,4,5]. The feeling of personal well-being was threatened. Epidemiological measures, although effective in preventing the spread of infectious diseases such as COVID-19, represent a serious threat to the mental health and well-being of the general population [6,7,8,9]. As far as individuals with serious mental illness were concerned, they were adversely affected by the COVID-19 epidemic. COVID-19-prevention measures, such as outbreak, isolation and quarantine, and social isolation, in these vulnerable individuals may have easily resulted in the development of fear and anxiety and thereby increase the incidence of stress-related diseases. Additionally, they can cause the exacerbation of pre-existing mental disorders in certain individuals [10]. Strict epidemiological measures represent social stressors that can easily trigger serious mental illnesses, such as depression and/or anxiety in previously healthy persons, and likewise contribute a larger burden to mentally ill ones [10].

An increase in the level of anxiety, depression, fear of loss, death, illness, stress, loneliness, post-traumatic symptoms and sleep disorders, and an increase in suicidality among both the general population and the population of patients with pre-existing mental illnesses were seen all around the world as a result of the COVID-19 pandemic [11,12,13,14,15,16].

The aforementioned psychological, psychiatric, and social problems caused by the circumstances accompanying the COVID-19 pandemic have previously been recognized in the literature as risk factors for the occurrence of suicidality [17,18]. People who already have a pre-existing mental illness, especially patients with affective disorders and those who react to stress with maladaptive patterns, are especially susceptible to suicidality [19].

Factors, such as the feeling of fear and stress caused by the COVID-19 pandemic and the level of social support, are singled out as those that play a leading role in the occurrence of psychiatric disorders during the COVID-19 pandemic [20,21,22]. Most of these mental health problems are recognized as leading risk factors for the occurrence of suicidality and suicidal behavior, especially in patients suffering from affective disorders such as unipolar depression and bipolar disorder or patients who have psychological disorders induced by stress [21,22]. Few studies have investigated the impact of the pandemic, especially on the suicidality and self-harm levels in people who were mentally ill prior to the pandemic onset, and the majority of them point out that health and social distress, fear and anxiety that emerged from the pandemic may be linked to an increase in suicidal risk and the severity of the psychopathological state in those individuals [23,24,25]. As far as we know, there have been no such investigations in Serbia. 

Therefore, this study aimed to compare the levels of fear and stress caused by circumstances related to the COVID-19 pandemic and levels of social support (exposure variables) in two groups of suicidal patients, grouped by the year of admission and who were diagnosed with a psychiatric disorder prior the pandemic onset. A comparison of suicidal patients during the first and second years of the COVID-19 pandemic according to exposure variables was carried out. In this way, an attempt was made to investigate the relationship between the exposure variables of fear, stress and social support and the outcome variable: suicidality.

Our initial hypothesis was that higher levels of stress and fear associated with the COVID-19 pandemic, as well as poor quality of social support, represent the factors that trigger existing mental illness conditions and contribute substantially to the occurrence of suicidality/self-harm thoughts and behavior. Furthermore, we assumed that the intensity of the stress and fear associated with the COVID-19 pandemic as well as social support in suicidal patients differ in relation to the period of the pandemic during which the suicidal relapse occurred.

## 2. Materials and Methods

This study was approved by the Ethical Committee of the Medical Faculty, University of Novi Sad (No. 12/21) and conducted according to the Declaration of Helsinki from 1975 (revised in 2013) [26]. This study is part of a wider research that will be the basis for writing a doctoral dissertation entitled “Frequency, characteristics of the clinical presentation and course of affective and stress-induced mental disorders during the SARS-CoV-2 virus pandemic” by Dr. Dušan Kuljančić.

### 2.1. Participants

This is an observational and retrospective–prospective study. The examined period covers two years during the COVID-19 pandemic in Serbia, from April 2020 to April 2022. This study included patients diagnosed with affective and stress-induced mental health disorders admitted to hospital treatment at the Psychiatry Clinic in Novi Sad, University Clinical Centre of Vojvodina due to suicide attempts and suicidal ideation. 

The patients included have been diagnosed with the mentioned diagnoses before the onset of the COVID-19 pandemic. During the pandemic, they relapsed. Suicidality/suicidal thoughts/self-harm/suicidal behavior was assessed through the standard Psychiatric Interview. 

To be eligible, participants needed to meet the following criteria: (1) diagnosis of affective disorder (F30–F39) or stress-induced disorder (F43.0–F43.9) according to ICD-10 criteria, (2) patients who had been diagnosed with the above-mentioned diagnoses earlier than the onset of the COVID-19 pandemic in Serbia, (3) patients who had relapsed during the pandemic and were admitted to hospital treatment due to suicidal behavior/thoughts, (4) absence of dual psychiatric diagnoses, (5) patients older than 18 years, and (6) patients who gave written consent to participate in this study. 

There were 108 patients who met the eligibility criteria. However, 8 of them were excluded because of incomplete medical documentation. Therefore, a total of 100 patients were finally included. In relation to the observed year in which they were admitted to the hospital, the patients were grouped into two subgroups: 2020–2021 and 2021–2022 and their data were compared. The subgroups of participants were equal, 50 patients in each, because in both observed years, there was an equal number of patients who were admitted to hospital treatment and met the criteria for inclusion in this study.

As for the participants’ biographical information, females were slightly more represented (55%) in relation to males (45%). The average age of the respondents was approximately 39 years. Most of the participants were unemployed and were educated at high school level.

Informed consent was obtained from all subjects involved in this study.

### 2.2. Instruments

During admission to hospital treatment, general sociodemographic data were collected (gender, age, education, marital status, number of children, number of members of the joint household, and work status). Additionally, as part of a standard psychiatric interview (which consists of questions about orientation, mood, anxiety, psychotic symptoms, obsessions and compulsions, dissociative symptoms, trauma and stress history, body image disturbances, eating disorders, sleep disturbances, somatic/pain disorders, suicidal and self-harm ideation/intentions/thoughts), patients filled in self-assessment data on the level of fear associated with the COVID-19 pandemic (the Fear of COVID-19 Scale, FCV-19S). Patients were asked to rate on a scale of 1 to 10 the extent to which the COVID-19 pandemic has been stressful for them in the past month. 

Furthermore, the participants were asked an explicit, closed type of question (yes/no/maybe) on whether they believe that the COVID-19 pandemic had a significant impact on the occurrence of their suicidal ideation. The majority of studies all around the world from the pandemic onset “emphasized the role of COVID-19 as a risk factor for higher levels of suicidal behaviors” [27,28,29,30]. Therefore, we assumed the beginning of the COVID-19 pandemic and all the accompanying unpleasant circumstances that led to a sudden change in the habits and lifestyles of individuals and entire populations as the main and only risk factor for the possible destabilization of mental health and the occurrence of suicidality in our patients. Hence, this clear and explicit question of whether patients believe that the COVID-19 pandemic has contributed significantly to the occurrence of suicidality among them was apt. 

To assess the level of social support, the Oslo Social Support Scale 3 (OSSS-3) was used at admission.

Well-trained psychiatry specialists who work at Clinic of Psychiatry, Clinical Centre of Vojvodina, participated in the admission of patients to the hospital and were responsible for data collection. At the time of data collection, at least two other medical staff members were present. Moreover, at least one non-psychiatrist in the liaison team was included in the consultation. To maintain data confidentiality, data accumulation and statistical analysis were performed by a different person than the psychiatrist who examined the patient [31]. Furthermore, reports from family or friends of participants were used to examine externally the validity of the self-reporting instruments [32]. Those were precautions taken to reduce the possible biases in diagnosis and information validity. The interviews were conducted in an interview room in an emergency ambulance, and there was no specific time limit [31]. 

#### 2.2.1. The Fear of COVID-19 Scale (FCV-19S)

The Fear of COVID-19 Scale is a seven-item unidimensional scale with robust psychometric properties. More specifically, reliability values such as internal consistency (α = 0.82) and test–retest reliability (ICC = 0.72) were acceptable. The total scores on the FCV-19S are comparable across both genders and all ages which suggest that it is a good psychometric instrument to be used in assessing and allaying fears of COVID-19 among individuals. Higher scores on the scale suggest a higher degree of fear of the COVID-19 pandemic and vice versa. We grouped the responses in a sitting manner—low (7–16), medium (17–25) and high levels of fear associated with the COVID-19 pandemic (26–35). The scale is available in the public domain [33]. 

#### 2.2.2. The Stress Measurement Instrument

A self-designed, self-reported Likert 1–10 scale was the instrument used to measure the personal perception of total stress amount connected with COVID-19 in respondents during the last month. The participants were asked one simple question “To what extent has the COVID-19 pandemic has STRESSFUL for you so far?”. Additionally, it has been explained what circumstances might be seen as stressful (There are several ways in which the coronavirus pandemic can cause stress in people, for example, the following situations: I got sick; I was admitted to hospital; Someone in my family got sick; Someone from my environment died as a result of being infected with the SARS-CoV-2 virus; I was/ I am in quarantine or self-isolation; I have been working from home; I have not been able to communicate with close and dear people, etc.) Participants had the opportunity to rate on a scale of 1 to 10 the extent to which the COVID-19 pandemic has been stressful for them so far (1 indicates the absence of stress, and 10 indicates intense, unbearable stress). The patient’s own perception of the degree of stress experienced during the pandemic and its impact on actual suicidal relapse of their condition was our main aim here. We would like to kindly draw your attention to the fact that we used this method of collecting data on the total amount of stress because it explicitly links stress symptoms to a specific circumstance whereas available general anxiety measures do not afford explicit links to specific circumstances such as the COVID-19 pandemic [20].

#### 2.2.3. The Oslo Social Support Scale 3 (OSSS-3)

The Oslo Social Support Scale (OSSS-3) is a 3-item self-reported measure of the level of social support. The OSSS-3 consists of three items which ask for the number of close confidants, the sense of concern from other people, and the relationship with neighbors with a focus on the accessibility of practical help. The sum score ranges from 3 to 14, with high values representing strong levels and low values representing poor levels of social support. The OSSS-3 sum score can be operationalized into three broad categories of social support: (a) 3–8 poor social support, (b) 9–11 moderate social support, and (c) 12–14 strong social support. For the OSSS-3, the internal consistency could be regarded as acceptable with α  =  0.640 and it can be considered as satisfying. The scale is available in the public domain [34,35]. 

### 2.3. Statistical Analysis

The SPSS for Windows 20 program was used for data processing, which works under the Microsoft Windows environment. The results are presented tabularly and graphically. Descriptive statistics are presented (frequencies and percentages for categorical data, as well as arithmetic means and standard deviations for quantitative data). In order to determine the relationship between suicidality in patients suffering from affective and stress-induced disorders in the first and second observed years, the χ^2^ test for categorical data was applied. For the clinical/social characteristics that are quantitatively expressed (stress, social support and fear), it was determined that the assumption of normality of the distribution of results was not fulfilled (results of the Kolmogorov–Smirnov test), and in order to analyze the differences between patients observed in the first and second pandemic year in terms of their expression, the non-parametric Mann–Whitney U test was applied. The correlation of stress, fear and social support with suicidality was examined by Spearman’s correlation coefficient.

## 3. Results

A total of 100 psychiatric patients suffering from affective and stress-induced mental health disorders were admitted for treatment at the Psychiatry Clinic of KCV Novi Sad in the period from April 2020 to April 2022 and participated in the research. Patients were divided into two groups of 50 each, depending on the year of the pandemic (2020–2021, 2021–2022).

### 3.1. Sociodemographic Characteristics of the Analyzed Sample

In terms of gender, at the level of the total sample, there were slightly more female respondents (55%) compared to male respondents (45%). Observed at the group level, there was a higher proportion of women in both years, but it is somewhat more pronounced in the first year. The average age of the respondents is approximately 39 years, and the standard deviation is 13. A total of 53% of the sample is younger than 40, and 47% are older. Respondents were somewhat older in the first pandemic year (average age 40.5, and 38 in the second). The sample of patients in relation to the age structure is largely uniform. Deviations occur only in the population of older working people aged from 40 to 60 years, where a difference in the number of suicides in our sample was recorded during the second pandemic year. The number of suicidal patients in this age category is almost 2.5-fold lower in the second pandemic year. The sociodemographic characteristics of our sample are listed in Table 1.

### 3.2. Self-Estimated COVID-19-Related Fear

In Table 2, it is suggested that over time, the mean value of reported fear intensity decreased compared to the first year. However, the data suggest that approximately half of the respondents in both observed years claim that they are afraid of dying from the virus, that the news about the virus makes them tense and nervous and that they cannot sleep because of worrying about the pandemic. In terms of the intensity of fear caused by the coronavirus epidemic, it is observed that the share of respondents who feel high levels of fear has decreased (from 42% in the first to 34% in the second year), as well as those who feel a medium intensity of fear (a third in the first and a fifth in the second observed year). A fifth of respondents felt a lower level of fear in the period 2020–2021 and almost half in the period 2021–2022.

The Mann–Whitney U test determined that there is no statistically significant difference between the two groups of patients in the average score on the scale of self-assessment of the intensity of fear caused by the coronavirus pandemic (*p* > 0.05). Table 2.

### 3.3. Self-Estimated COVID-19-Related Stress and its Relation to Present Mental Health Problems

Table 3 shows that there was no statistically significant difference between the two groups of patients in the average score on the stress scale (*p* > 0.05). However, the share of those who assessed stress with the highest scores is approximately equal in both groups.

Table 4 shows that there is a statistically significant relationship between the individual perception of the impact of the COVID-19 pandemic on the occurrence of mental health problems and suicidality and the period 2020–2021 of the pandemic χ^2^(2, N = 100) = 8.347, *p* = 0.015. The obtained findings indicate that the percentage of patients who do not think that the coronavirus pandemic had an impact on the occurrence of psychological problems is significantly higher in the second examined year compared to the first. On the other hand, the percentage of respondents who stated “maybe” is higher in the first year of the pandemic. The calculated value of the indicator Cramer’s V = 0.289 tells us that there is a moderately strong relationship between the variables.

### 3.4. The Social Support Level and its Relation to Present Mental Health Problems

The Mann–Whitney U test determined that there was no statistically significant difference between the two groups of patients in the average score on the social support scale (*p* > 0.05) Table 5.

Table 6 and Table 7 list the correlation coefficients between suicidal behavior, stress intensity and scores on the scales of social support and fear intensity, separately for the two observed periods.

Looking at Table 6 and Table 7, it can be seen that no statistically significant correlation was found between suicidality, stress intensity, fear and the score on the social support scale in both examined periods (*p* > 0.05).

## 4. Discussion

This study attempted to shed light on the role of levels of social support, fear and stress related to the COVID-19 pandemic in the development of suicidality in patients with affective and stress-induced psychological disorders who relapsed and were admitted to hospital for suicidality as a prominent symptom during the COVID-19 pandemic, comparing exposure variables among two groups of patients admitted in 2020 and 2022.

The results of our study did not fully confirm the initial hypothesis that higher levels of stress, the fear associated with the COVID-19 pandemic as well as lower levels of social support are triggers for relapse in psychiatric patients during the pandemic. Only the subjective perception of distress related to the pandemic was statistically significantly associated with the development of suicidality in our patients. Additionally, the levels of stress, fear and social support differ among the participants in this study in relation to the examined year, as we assumed.

The COVID-19 pandemic has increased the risk of worsened mental health among people with mental health disorders. This worsening has been related to the symptoms of the infection but also to other preventable social circumstances, such as the loss of therapeutic interventions, the loss of employment, and low financial income [36,37]. The research shows that the increase in suicide rate in psychiatric patients during public health crises, such as pandemics, is connected with the fear of getting sick, becoming a burden to the family and a fear for life, anxiety, social isolation and distress [38]. It is well documented that the social support level in psychiatric patients plays a significant role in maintaining remission and is inversely related to suicidal relapse [20,39].

The feeling of fear of COVID-19, especially the fear of getting sick and dying, is significantly associated with a perceived level of distress [37,38,39,40]. The lowest and highest scores on the FCV-19S scale are recorded in East Asia and Spain (16 +/− 6; 18 +/− 5) and Australia (19 +/− 6), with a special risk for people who are younger, female, unemployed and students, as well as health care workers, while older people, men and those with better education showed lower values on the Fear of COVID-19 Scale [40,41,42]. The data on the level of fear in previously mentally ill people show clearly that it is significantly higher than in the general population. Furthermore, it seems more severe in nature and etiology compared to people without mental disorders and even in relation to those people who have severe somatic comorbid diseases and who represent a risk group for contracting COVID-19 [43]. Namely, the fear shown by the mentally ill in relation to the COVID-19 pandemic is more pathological and represents the combination of distress and increased levels of depression and anxiety, which together can lead to suicidality [11,43].

The studies suggest that the level of fear connected with COVID-19 solely cannot be seen as a sufficient trigger to develop suicidality relapses in the mentally ill. The wider context of the population and the fear intensity should be considered with suicidality occurrence. Namely, at the very beginning of the pandemic in our region, especially in Croatia, there was an increase in the rate of suicides, which can be connected to the collective psychological tension created in the society—the fact that something new and unknown and life-threatening is emerging accompanied by the uncertainty surrounding its origin, way of spreading, health consequences and the socioeconomic and financial consequences of the pandemic [44,45]. The fear generated by the objective circumstances of the beginning of the pandemic is interpreted as a trigger for numerous suicide attempts at the beginning of the pandemic in Croatia, and we have similar observations in relation to Serbia too [10,11,45]. However, on the other hand, a study covering 21 countries worldwide found that suicide rates initially decreased during the pandemic regardless of the stressful and fearful circumstances accompanying the onset of the pandemic [46]. This difference in findings can be attributed to the difference in the sample because only economically developed and stable countries were included, unlike Serbia and its immediate surroundings, which are still developing countries; this assumes that economic and material stability and security play a significant role in mental stability [47,48,49,50]. 

A statistically significant association between the level of fear associated with the COVID-19 pandemic and suicidality was not found in our sample. It can be assumed that the reason for this result is that in our general population, there were previously many stressful situations, which in terms of intensity and duration, far exceed the COVID-19 pandemic, such as the bloody disintegration of Yugoslavia, numerous fratricidal wars, hyperinflation, poverty, misery, harsh transition and unemployment [16,51]. Therefore, mentally ill people seem to be more resilient to the current pandemic and its negative impact on mental health. However, we cannot minimize the negative impact of epidemiological measures and the state of emergency on mental health because the rates of occurrence of psychopathological phenomena are among the highest in the world [6,10,11,43,52,53].

On the other hand, regardless of any potential resilience, the respondents involved in this study perceive the COVID-19 pandemic as an etiological factor that significantly contributed to the current deterioration of their mental state.

The subjective feeling of stress related to the COVID-19 pandemic in our sample of patients is very pronounced. The feeling of personal powerlessness, helplessness, risk of getting sick, a threat to life and well-being, chronic economic instability and constant potential instability in the region are all socioeconomic and demographic factors that are the source of chronic stress, which in our general population during the pandemic, was at a much higher level than in other countries of the world [52,54]. Therefore, even patients who were being treated for psychiatric disorders could not be spared this level of distress, although we did not find a statistically significant connection between the level of stress and suicidality. However, we have yet to witness the effects of cumulative stress on mental health and suicide rates [55,56].

Finally, an equally important factor for mental health is adequate social support. Through the results of our research, for the appearance of negative outcomes, especially in the field of mental health after public health crises, the time during which stressors act is crucial. Namely, as a protective factor against psychological distress, our data also support adequate social support, which is in line with the conclusions of many international studies [57,58]. In the second examined year, patients without a single close person to rely on dominated. Only then did the burden of cumulative distress come to the fore. Furthermore, social support was inadequate and deficient, so it can be assumed that the combined distress factors contributed to suicidality. Therefore, social support plays a significant role in the relapse of mental health problems in the public health crisis environment. Good social support, as well as individual perception of it, is a protective factor for the development of anxiety and depression in vulnerable individuals in crisis situations, and therefore, reduces the frequency of known risk factors for suicidality, accordingly to our results as well as the other studies worldwide [58,59]. 

However, our results show total scores on the scale of social support among both of our examined groups which are relatively high and equal. Therefore, it is necessary to consider overall social support and “forced closeness” in people with whom true closeness is not so pronounced, and those who were forced to stay during the “lockdown” measures [59]. 

Our study has numerous positive aspects and strengths. Namely, it is one of the few in this area, as far as we know, that deals with the topic of mental health problems in previously mentally ill people during the pandemic. An advantage of this study is the time the data were collected during the COVID-19 pandemic, i.e., monitoring the connection of the examined variables in relation to the actual course of the pandemic in Serbia.

On the other hand, we are also aware of numerous deficiencies and shortcomings of this study. First, the very design of the research as an observational study of a retrospective–perspective character does not allow the determination of causality between the examined phenomena. To overcome this problem and to better understand casual relationships, a longitudinal study is needed to validate our results. Furthermore, the sample size is not sufficient to characterize the sample as representative. It is necessary to conduct a similar study in cooperation with several different psychiatric institutions throughout Serbia to overcome this problem. Additionally, to draw clearer connections about the impact of the mentioned factors on suicidality, we believe it is necessary to design a study where only the first manifestations of mental illnesses would be included. However, sample sourcing and resources are again the problems. Finally, in the process of data collection, a self-reporting bias may be present because a self-reporting questionnaire was used [60].

## 5. Conclusions

The purpose of the present observational study was to investigate the relationship between fear and stress caused by the circumstances related to the COVID-19 pandemic and levels of social support, and suicidality in patients diagnosed with affective and stress-induced disorders already admitted to a psychiatric clinic for attempted suicide or suicidal ideation in the periods 2020–2021 and 2021–2022 of the pandemic. This study observed a significant relationship between distress caused by COVID-19 and suicidality in the first period of the pandemic (April 2020–April 2022); in contrast, no positive correlations were found between suicidal behavior, stress intensity, fear and perceived social support. The importance of adequate social support in maintaining remission in examined psychiatric patients during the COVID-19 pandemic was confirmed. In the second examined year of the pandemic, patients who had insufficient social support dominated, and the level of accumulated stress came to the fore to the largest possible extent. As far as the implications of this study are concerned, in future public health crises, it will be necessary to pay more attention to the mental health of both the general population and already mentally ill patients, especially in developing countries and in countries with many socioeconomic conflicts. In such situations, the health care system should not neglect psychiatric patients and their well-being, because exposure to stressors and inadequate help during that period are shown to be indicators of delayed relapses of mental disorders. Therefore, decision makers should be aware of this and prevent the subsequent increase in mental disorders. This study raises new questions and encourages new, more detailed research, especially to clarify the nature of the relationship between both the quantity and quality of social support and the positive mediation of the impact of stressors in public health crises.

## Figures and Tables

**Table 1 brainsci-13-00812-t001:** Sociodemographic characteristics of the entire sample (N = 100) and according to the year of the pandemic.

Variables	Whole Sample (N = 100)	Year	
		2020–2021 (N = 50)	2021–2022 (N = 50)
Sex			
Male	45 (45%)	21 (42%)	24 (48%)
Female	55 (55%)	29 (58%)	26 (52%)
Age			
M ± SD	39.28 ± 13.02	40.58 ± 12.68	37.98 ± 13.36
Less than 40 years	53 (53%)	24 (48%)	29 (58%)
40–59 years	20 (20%)	14 (28%)	6 (12%)
60–74 years	25 (25%)	12 (24%)	13 (26%)
More than 75 years	2 (2%)	0 (0%)	2 (4%)
Education status			
Unfinished elementary school	9 (9%)	2 (4%)	7 (14%)
Finished elementary school	25 (25%)	12 (24%)	13 (26%)
High school	53 (53%)	28 (56%)	25 (50%)
College and university	13 (13%)	8 (16%)	5 (10%)
Employment status			
Unemployed	45 (45%)	22 (44%)	23 (46%)
Employed	32 (32%)	15 (30%)	17 (34%)
Retired	6 (6%)	3 (6%)	3 (6%)
Housewife	8 (8%)	6 (12%)	2 (4%)
Student	9 (9%)	4 (8%)	5 (10%)

Notes: M—mean; SD—standard deviation.

**Table 2 brainsci-13-00812-t002:** The Mann–Whitney U test for examining differences between two groups of patients (hospitalized during the first and second pandemic year) regarding the average score on the self-report scale of fear intensity.

Year	N	Mean Rank	Mann–Whitney U	Wilcoxon W	Z	*p*
2020–2021	50	55.67	991.500	2266.500	−1.785	0.074
2021–2022	50	45.33				

Note: N—number of respondents, Mean Rank—average rank, Mann–Whitney U—test value, Wilcoxon W—statistic, Z—standardized statistic, and *p*—statistical significance.

**Table 3 brainsci-13-00812-t003:** The Mann–Whitney U test for examining differences between two groups of patients (hospitalized during the first and second pandemic years) regarding the average score on the stress scale.

Year	N	Mean Rank	Mann–Whitney U	Wilcoxon W	Z	*p*
2020–2021	50	49.38	1194.000	2469.000	−0.389	0.697
2021–2022	50	51.62				

Note: N—number of respondents, Mean Rank—average rank, Mann–Whitney U—test value, Wilcoxon W—statistic, Z—standardized statistic, and *p*—statistical significance.

**Table 4 brainsci-13-00812-t004:** The χ^2^ test of correlation between the impact of coronavirus on the occurrence of mental disorders and the year of the pandemic.

Group	N	Do You Think That the Coronavirus Pandemic Had a Significant Impact on Suicidality Lately?			χ^2^	*p*
		Yes	No	Maybe		
2020–2021	50	30	14	6	8.347	0.015
2021–2022	50	27	23	/		
Total	100	57	37	6		

Note: N—number of respondents, χ^2^—statistic, and *p*—statistical significance.

**Table 5 brainsci-13-00812-t005:** The Mann–Whitney U test for examining differences between two groups of patients (hospitalized during the first and second pandemic year) in terms of the total average score on the social support scale.

Year	N	Mean Rank	Mann–Whitney U	Wilcoxon W	Z	*p*
2020–2021	50	51.15	1217.500	2492.500	−0.225	0.822
2021–2022	50	49.85				

**Table 6 brainsci-13-00812-t006:** Correlation between suicidality assessed through the standard psychiatric interview, stress intensity and scores on the scales of social support and fear intensity in the period 2020–2021.

2020–2021			Suicidal Thoughts	Social Support	Stress	Fear
Spearman’s rho	Suicidal thoughts	Correlation Coefficient	1.000	−0.124	0.055	0.153
		Sig. (2-tailed)		0.391 *	0.707 *	0.290 *
		N		50	50	50
	Social support	Correlation Coefficient		1.000	−0.096	−0.107
		Sig. (2-tailed)			0.506 *	0.458 *
		N			50	50
	Stress	Correlation Coefficient			1.000	−0.046
		Sig. (2-tailed)				0.753 *
		N				50
	Fear	Correlation Coefficient				1.000
		Sig. (2-tailed)				
		N				

* correlation is significant at the 0.05 level.

**Table 7 brainsci-13-00812-t007:** Correlation between suicidal behavior, stress intensity and scores on the scales of social support and fear intensity in the period 2021–2022.

2021–2022			Suicidal Thoughts	Social Support	Stress	Fear
Spearman’s rho	Suicidal thoughts	Correlation Coefficient	1	−0.166	−0.187	0.060
		Sig. (2-tailed)		0.248 *	0.194 *	0.679 *
		N		50	50	50
	Social support	Correlation Coefficient		1	0.240	−0.004
		Sig. (2-tailed)			0.093 *	0.979 *
		N			50	50
	Stress	Correlation Coefficient			1	−0.103
		Sig. (2-tailed)				0.476 *
		N				50
	Fear	Correlation Coefficient				1
		Sig. (2-tailed)				
		N				

* correlation is significant at the 0.05 level.

## Data Availability

The data that support the findings of this study are available from the corresponding author (D.K.) upon reasonable request.

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
