# Peer review of "The Role of COVID-19-Associated Fear, Stress and Level of Social Support in Development of Suicidality in Patients Diagnosed with Affective and Stress-Induced Psychiatric Disorders during the COVID-19 Pandemic—A Comparative Analysis"

_brainsci, 2023, doi:10.3390/brainsci13050812_

Round 1

Reviewer 1 Report

Comments and Suggestions for Authors

Thank you for the opportunity to review the article titled - The Role of COVID-19-associated Fear, Stress and Level of Social Support in Development of Suicidality in Patients Diagnosed with Affective and Stress-Induced Psychiatric Disorders During the COVID-19 Pandemic.   I appreciate the efforts of the authors; however, I have a few suggestions to make to improve the quality and readability of the article.

Introduction

Lines- 44-46 – need to rewrite for clarity

Lines 65 – 70 need to be rewritten as the aims are unclear to the readers- You need to split the sentence into at least two. – are the authors exploring the stressors to trigger existing mental illness conditions or newly diagnosed patients with emotional disorders?

Grammatical errors and spelling mistakes are found., e.g. “Befor” in line 84

In the inclusion criteria, it is unclear whether they already had a psychiatric diagnosis before covid-19.

The sampling needs to be clarified- did the authors include all the 100 patients who fulfilled the criteria who accessed the health services in two years? Then how come precisely 50 in the first year and another 50 in the second year? Additionally, how did they get the exact 100- did the authors keep 100 as their sampling size prior to the study?

Who collected data, and what was their qualification, as they dealt with a highly vulnerable group? What precautions were taken to reduce the biases- self-reporting bias and social desirability bias?

The statistical analysis part can be summarised.

Lines- 146- 147- Standard deviation found to be very high; therefore, the patients categorising under the age quartile would give a better understanding of the population. Below 40 and above 40 may not be a good way of dividing the population, as – National suicide statics could help bin them.

The analysis is not in line with the title of the manuscript or the aim of the study. It needs to be clarified why the authors have given too much thrust to comparing the two years. Instead, consider the outcome variables such as suicidality and exposure variables of fear, stress and social support. Therefore, I suggest redoing the first table to align the results with the aim of the manuscript.

Table 2- is inappropriate- readers may be interested in whether the difference is more significant than percentages. Since the scale is standardised, the computer scores would have allowed the authors to have higher-level statistics and omitted this table.

The authors have claimed their fear was reduced (Lines 159-162); however, the clinic also had 50 cases in the second year. Therefore, their claim is not justified.

And in the third table, the authors have said there is no difference (171- 173). It also needs to be clarified why the authors have used non-parametric tests such as Mann- Whitney U while the data parameters are well defined.

Table 5 tells the chi-square test of correlation- but the purpose of the chi-square test is different – I would suggest the authors get an expert opinion of statisticians.

In the analysis part, the third-person narration would be ideal.

Chart 1 and Chart 2 communicate little, and they take up much space, which is not a good idea. Therefore, this data can be presented in some other form. I suggest a complex chart with the significant variables rather than the year. Need to remove it. I don’t understand why the authors gave too much importance to the year- instead, they go for major variables mentioned in the title and aims of the paper.

Too many tables – please rework it.

Discussion

Lines 232 to 235- I think the authors should have had more consideration for these original aims of the study while doing their analysis.

The discussion needs to be summarised, trimmed, and thoroughly edited.  

-          Need to explain usual stressors in already diagnosed patients in relapse and suicidality

-          Secondly, what the covid specifically trigger?  What is the professional and practical implication of this study?

The conclusion part is also too much. Trim it, and we don’t expect any references in the conclusions-

Thanks again 

Comments on the Quality of English Language

Authors need to improve the English language to communicate with the readers. 

Author Response

Reply to the Review Report No. 1

Dear Reviewer,

Firstly, I am glad to hear your suggestions to  help improve our work. As requested, I will now respond step by step to each one.

  1. „Lines- 44-46 – need to rewrite for clarity.“- It has been rewritten and expanded with more detail information and a new reference.”

  1. „Lines 65 – 70 need to be rewritten as the aims are unclear to the readers- You need to split the sentence into at least two. – are the authors exploring the stressors to trigger existing mental illness conditions or newly diagnosed patients with emotional disorders?“- This section is rewritten and paraphrased to achieve the clarity. We are exploring the stressors to the pre-pandemicly-existing mental illness conditions- affective and stress-induced mental disorders who relapsed during the pandemic.

  1. „Grammatical errors and spelling mistakes are found., e.g. “Befor” in line 84“- This error is corrected and the whole manuscript has undergone the professional proofreading to improve English language quality.

  1. „In the inclusion criteria, it is unclear whether they already had a psychiatric diagnosis before covid-19.“- The second Reviewer had the similar notice here as well. I tried to write clearly this time what were the criteria for inclusion in the research. I repeat once again that only those patients who had diagnosed psychiatric disorders before the pandemic were included. During the pandemic, those patients only relapsed with suicidality as the main feature of the clinical picture.
  2. „The sampling needs to be clarified- did the authors include all the 100 patients who fulfilled the criteria who accessed the health services in two years? Then how come precisely 50 in the first year and another 50 in the second year? Additionally, how did they get the exact 100- did the authors keep 100 as their sampling size prior to the study?“- There were 108 patients who met the eligibility criteria. However, 8 of them were excluded because of the incomplete medical documentation. So total of 100 patients were finally included. In relation to the observed year in which they were admitted within the hospital, the patients were grouped into two subgroups: 2020-2021 and 2021-2022 pandemic year and their data were compared. The subgroups of participants were equal, 50 each, because in both observed years there was an equal number of patients who were admitted to hospital treatment and met the criteria for inclusion in the study.

  1. „Who collected data, and what was their qualification, as they dealt with a highly vulnerable group? What precautions were taken to reduce the biases- self-reporting bias and social desirability bias?“- Well-trained psychiatry specialists who work at Clinic of Psychiatry, Clinical Centre of Vojvodina, participated in the admission of patients to the hospital were responsible for data collection. At time of data collection at least two other medical staff members were present. Moreover, at least one non-psychiatrist in liaison team was included in the consultation. To maintain the data confidentiality, data accumulation and statistical analysis were performed by a different person than the psychiatrist who examined the patients. Furthermore, reports from family or friends of participants were used to examine externally the validity of the self-reporting instruments. Those were precautions taken to reduce the possible biases in diagnosis and information validity.

  1. „Lines- 146- 147- Standard deviation found to be very high; therefore, the patients categorising under the age quartile would give a better understanding of the population. Below 40 and above 40 may not be a good way of dividing the population, as – National suicide statics could help bin them.“- I agree. We corrected the first table accordingly. According to the Data of the Republic Institute for Statistics of Serbia, we took a good limit of 40 years. Namely, of the total number of suicides in Serbia in 2021, only 13% are committed by people under the age of 40. All other suicides are committed by people over 40 years old. That is why that population is divided into three subclasses – from 41 to 60 years old, from 61 to 74 years old and over 75 years old.

  1. „The analysis is not in line with the title of the manuscript or the aim of the study. It needs to be clarified why the authors have given too much thrust to comparing the two years. Instead, consider the outcome variables such as suicidality and exposure variables of fear, stress and social support. Therefore, I suggest redoing the first table to align the results with the aim of the manuscript.“- The title and the aim of the manuscript has been modified to suit better the statistical analysis. According to your instructions, however, we have decided to align the title and objective of the work with current statistics. We were not able to change the entire statistical analysis in such a short time according to what you requested, even though you said that in that case your decision might not be favorable. And on the other hand, we were in a very uncomfortable position since the second reviewer thought that the current statistical analysis was adequate and had no major objections. He stated that the statistical part was cited as "well written".
  2. „Table 2- is inappropriate- readers may be interested in whether the difference is more significant than percentages. Since the scale is standardised, the computer scores would have allowed the authors to have higher-level statistics and omitted this table.“- We have avoided Table 2 to trim the results section.

  1. „The authors have claimed their fear was reduced (Lines 159-162); however, the clinic also had 50 cases in the second year. Therefore, their claim is not justified.“- We think the reviewer is contradictory here. Namely, he decisively stated that if we were to change the entire statistical analysis, descriptive statistics would be sufficient. And then he makes the claim that such a claim needs to be checked with some statistical tests. We were really in a dilemma as to what to do, and as we are aware that the Mean value cannot be taken as evidence for a claim, we decided to correct the claim itself. Another reviewer gave us a similar recommendation. That's why we put that sentence in possible and not certain terms. We said "It suggests that the mean value of reported fear intensity is lower compared to the first year.”

  1. “And in the third table, the authors have said there is no difference (171- 173). It also needs to be clarified why the authors have used non-parametric tests such as Mann- Whitney U while the data parameters are well defined.“- The data parameters are well defined, yes. However, when we have done the normality test, the assumption of normality of the results was not satisfied. Normality tests are done to see what the distribution is like, so if it deviates from normal, non-parametric tests that do not set strict conditions are applied. Ofcourse yes their strength is less. Some tests are "robust" enough to be tamper-resistant normality. If you think it is necessary to transform the data, we did not have enough provided time.

  1. “Table 5 tells the chi-square test of correlation- but the purpose of the chi-square test is different – I would suggest the authors get an expert opinion of statisticians.“, „If you continue to claim so, I don’t comment on it anymore. However, since more appropriate tests are available, why should the authors stick to them?“- We contacted a statistician who believes that there is nothing controversial about the choice of method. The choice of statistical method is adequate according to the statistician's opinion and is related to the reply on the previous comment.

  1. „In the analysis part, the third-person narration would be ideal.“- The manuscript went through english language grammatic and style editing and we have corrected as much as we could accordinglly.

  1. „Chart 1 and Chart 2 communicate little, and they take up much space, which is not a good idea. Therefore, this data can be presented in some other form. I suggest a complex chart with the significant variables rather than the year. Need to remove it. I don’t understand why the authors gave too much importance to the year- instead, they go for major variables mentioned in the title and aims of the paper.“- We have avoided Chart 1 and 2 to trim the results section.

  1. Discussion

Lines 232 to 235- I think the authors should have had more consideration for these original aims of the study while doing their analysis.- We have clarified and made the aims more precise to be in line with the analysis done.

The discussion section is summarised, trimmed, and edited.  

We are grateful to the reviewer for the questions with which he directed us in the direction in which we should write the discussion. We tried our best to answer his questions. Hopefully the discussion is now improved.

  1. Conclusion

The conclusion part is trimmed too, as you suggested. The reference is avoided.

Once again, I would like to emphasize that we are grateful for your suggestions. We regret that we were not able to implement them all because there was not enough time due to the revision deadline. Also, we found ourselves in an unenviable position since we received contradictory recommendations both from you in different parts of the work and from another reviewer. So we didn't know what to do, because if we followed your recommendations we would violate all the recommendations of the other reviewer and vice versa. We tried to find a middle ground.

We hope that the work will still receive a positive evaluation.

Respectfully,

Dušan Kuljančić

Reviewer 2 Report

Comments and Suggestions for Authors

The Role of COVID-19-associated Fear, Stress and Level of Social Support in Development of Suicidality in Patients Diagnosed with Affective and Stress-Induced Psychiatric Disorders During the COVID-19 Pandemic

Summary of the research

The purpose of the present observational study was to investigate the relationship between fear and stress caused by the circumstances related to the COVID-19 pandemic and levels of social support, suicidality in patients diagnosed with affective and stress-induced disorders already admitted to a psychiatric clinic for attempted suicide or suicidal ideation. The study observed a significant relationship between distress caused by COVID-19 and suicidality in the pandemic year (April 2020); in contrast, no positive correlations were found between suicidal behavior, stress intensity, fear and perceived social support.

Abstract

The study abstract appears well structured and offers the reader a clear idea of the article content, and the number of words is suitable with the number required by the journal. It might be useful to include within the abstract the names of the questionnaires used for the surveys conducted and the number of participants involved.

Keywords

The keywords are adequate regarding the correct number of keywords required by the journal. They are also relevant with the topics covered in this article.

Introduction

The section appears generally well-structured, however, there are a couple of minor changes that are recommended:

- At line 42 it would be appropriate to cite a bibliographic reference.

- At line 64 it would be appropriate to cite the studies mentioned regarding the impact of the pandemic on suicidality in persons previously affected by psychiatric disorders.

Finally, it would be appropriate for the authors to indicate the starting hypothesis at the conclusion of the introductory section.

Materials and Methods

In this section the authors describe in detail the methodology used to conduct the study and the material used. This section has some limitations:

2.1 Participants

-          Line 84: there is a typo in the word "befor" (before).

-          Line 85: there is a typo in the word "rellapsed" (relapsed)

-          How come in the ICD-10 criteria the authors inserted ellipses in reference to the included diagnoses? It is unclear whether by F3 they refer to all affective disorders (F30 to F39), and it is also unclear whether in the second parenthesis (F43...) the dots indicate additional diagnoses beyond the one actually indicated by the acronym F43 i.e., "Severe stress reaction and adjustment disorders." Moreover, the disorders following this one are very different in terms of symptoms, emotions and behavior (F44 "Dissociative disorders," F45 "Somatoform disorders" etc.).

-          It would be appropriate to include the participants' biographical information (age, gender and also social status, since the authors also investigated social support among the variables) in this section and not within the Results section.

-          It is unclear what the authors mean by "same characteristics" in line 91: in addition to the diagnosis, it is necessary to specify which characteristics were common to the selected patients.

-          It is unclear how the division into the two groups of patients of 50 each was made: based on the year of symptom onset or admission within the facility?

2.2 Instruments

-          Line 98: It would be appropriate to specify which areas are investigated in the regular psychiatric interview.

-          It is necessary for the authors to include the full name of the FCV-19S questionnaire before mentioning only the abbreviation (line 100).

-          Is the Likert 1-10 scale for measuring stress the only instrument used by the authors to measure this variable? In addition, it would be appropriate to include within this paragraph a subsection (as was done for the FVC-19S and OSS-3) to more extensively explicate how stress is measured.

-          It would be appropriate for the authors to specify in this paragraph that the question about belief in the influence of COVID-19 on suicidal ideation be a closed question (yes/no/maybe) since it is only understood by reading the results later. In any case, it seems quite reductive to ask such an important question in closed mode, considering the fact that suicidal ideation may have a cause-and-effect correlation. It would be important for the authors to argue and justify the decision to detect this information in this mode.

Results

In the Results section, the authors present the findings of their research. This part is well written, but we suggest the following modifications:

-          It is not clear whether the results within Table 2 are related to the FCV-19S questionnaire; it would be appropriate to specify this.

-          It would be appropriate to specify in Table 7, regarding correlations between instruments, whether suicidal behavior indicates the survey done by closed-ended "yes/no/maybe" response or refers to another assessment (e.g., the Psychiatric Interview).

-          Line 155-156, it would be more appropriate to put the sentence in possible and not certain terms (For example, one could change "it means" to "it suggests"

Discussion

In this section, the authors discuss the research results more in depth. The limitations and the implications of the study are presented.

There are some modifications that we suggest:

-          Line 237: a bibliographic reference is missing to support the sentence "The feeling of fear of COVID-19 is an inevitable companion of stress."

-          Line 255: it would be appropriate to report the data for clarity in the text.

-          Line 262: it would be appropriate to include a reference regarding the increased suicide rate in Croatia during the early period of the pandemic.

-          It would be appropriate to include a literature reference inherent in the link between socioeconomic status, being a resident of developing countries, and mental well-being.

-          Line 274: there is a typo, one point too many after the date '2021'

-          Line 283: there is a missing bibliographic reference to support the sentence.

-          Line 289: there is a typo in the word 'menatlly' (mentally).

-          Line 338-339: cite studies to support the sentence "the feeling of fear, which, as is well known, has - a particularly pronounced social component, is contagious."

Conclusions

The section appears generally well-structured and provides a comprehensive overview of the issues discussed in the paper.

Use of English

The English language should be revised by a mother tong copyeditor.

Comments on the Quality of English Language

The article needs major revision

Author Response

Reply to the Review Report No. 2

Dear Reviewer,

Firstly, I am glad to hear your comments on our work. As requested, I will now respond step by step to each one.

Abstract

  1. I have rewritten the abstract in order to include the names of the questionnaires used for the survey conducted and the number of participants involved, as you have suggested.

Introduction

  1. „At line 42 it would be appropriate to cite a bibliographic reference.“- I have included the cite of 3 relevant references.

  1. „At line 64 it would be appropriate to cite the studies mentioned regarding the impact of the pandemic on suicidality in persons previously affected by psychiatric disorders.“- I have cited the mentioned studies, and also made a additional comment on those findings.

  1. „Finally, it would be appropriate for the authors to indicate the starting hypothesis at the conclusion of the introductory section.“- The starting hypothesis has been added.

Materials and Methods

Participants

  1. „Line 84: there is a typo in the word "befor" (before).“- The typo is corrected.

  1. „Line 85: there is a typo in the word "rellapsed" (relapsed).“- The typo is corrected.

  1. „How come in the ICD-10 criteria the authors inserted ellipses in reference to the included diagnoses? It is unclear whether by F3 they refer to all affective disorders (F30 to F39), and it is also unclear whether in the second parenthesis (F43...) the dots indicate additional diagnoses beyond the one actually indicated by the acronym F43 i.e., "Severe stress reaction and adjustment disorders." Moreover, the disorders following this one are very different in terms of symptoms, emotions and behavior (F44 "Dissociative disorders," F45 "Somatoform disorders" etc.).“- We agree that given that way the inclusion criteria are not clear and not precise. By F3... we wanted to say that we involved all affective disorders. However, it is not clear that way so we changed it as you suggested (F30-F39). The same situation is with the DG F43. The dots do not indicate additional diagnoses following in the F4 section. To make it cealr we have putted it like this- F43.0-F43.9.

  1. „It would be appropriate to include the participants' biographical information (age, gender and also social status, since the authors also investigated social support among the variables) in this section and not within the Results section.“- We have agreed to take your advice on this issue. The biographical information are now placed within the „Participants“ subsection.

  1. „It is unclear what the authors mean by "same characteristics" in line 91: in addition to the diagnosis, it is necessary to specify which characteristics were common to the selected patients.“- The first reviewer aslo reffered to unclarity of the inclusion criteria so we have rewritten them in order to make them more accurate and precise. I hope that way we have achieved it.

  1. „It is unclear how the division into the two groups of patients of 50 each was made: based on the year of symptom onset or admission within the facility?”- The division was made based on the year of admission within the hospital. It has been clearly stated in the manuscript now.

Instruments

  1. „Line 98: It would be appropriate to specify which areas are investigated in the regular psychiatric interview.“- We have taken this advice of yours and have specified the areas which have been investigated in the regular psychiatric interview.

  1. „It is necessary for the authors to include the full name of the FCV-19S questionnaire before mentioning only the abbreviation (line 100).“- It has been corrected.

  1. „Is the Likert 1-10 scale for measuring stress the only instrument used by the authors to measure this variable? In addition, it would be appropriate to include within this paragraph a subsection (as was done for the FVC-19S and OSS-3) to more extensively explicate how stress is measured.”- Yes, the self-designed, self-reported Likert 1-10 scale was the only instrument used to measure stress in our respondents. At the very beginning of the pandemic, when we started collecting this data, there were no questionnaires that were specialized and created specifically for the stress caused by the COVID-19 pandemic. There are examples of studies around the world, especially Chinese authors from the beginning of the pandemic, who used similar measurement instruments when analyzing the stress caused by the pandemic [Zhang Y et. all. 2021].

Given the circumstances, that we started collecting data at the very beginning of the pandemic and that the data were collected at the time of admission of relapsing, suicidal patients, we  decided to use the simplest, shortest instruments. We agree, as I hope, we were dealing with a vulnerable, specific population at a difficult time in their relapsing disease and that that is the main reason why we needed an easy-to-use instrument. Precision and explicitness in asking questions to the participants in this study in a limited time allowed us to obtain essential data to link the suicidality in our patients and circumstances of the pandemic. The patient's own perception of the degree of stress experienced during the pandemic and it՚s impact on actual suicidal relaps of their condition was our main aim here.

I agree that it would be appropriate to include within this paragraph a subsection (as was done for the FVC-19S and OSS-3) to more extensively explicate how stress is measured, and it is done.

  1. „It would be appropriate for the authors to specify in this paragraph that the question about belief in the influence of COVID-19 on suicidal ideation be a closed question (yes/no/maybe) since it is only understood by reading the results later. In any case, it seems quite reductive to ask such an important question in closed mode, considering the fact that suicidal ideation may have a cause-and-effect correlation. It would be important for the authors to argue and justify the decision to detect this information in this mode.“- We made clear what was the exact type of question been asked and gave an extensive explanation for our choise based on bibliographic citations.

Taking into account the conclusions of studies around the world from the beginning of the pandemic which "emphasized the role of COVID-19 as a risk factor for higher levels of suicidal behaviors", when we started our study in 2020, we just went in same direction. Therefore, we assumed the beginning of the COVID-19 pandemic and all the accompanying unpleasant circumstances that led to a sudden change in the habits and lifestyles of individuals and entire populations as the main and only risk factor for the possible destabilization of mental health and the occurrence of suicidality in our patients. Hence the clear and explicit question of whether patients believe that the COVID-19 pandemic has contributed significantly to the occurrence of suicidality among them. Although we were also aware that such an approach could be reductive. But we still accepted that risk, guided by the conclusions of world studies, but also with the intention of focusing really only on the COVID-19 pandemic as risk factor for occurrence of suicidality.

Results

Since, the first reviewer had more comments on the presentation of the results.  I tried to fullfil both suggestions.

  1. „It is not clear whether the results within Table 2 are related to the FCV-19S questionnaire; it would be appropriate to specify this.“- I specified this in the text.

  1. „It would be appropriate to specify in Table 7, regarding correlations between instruments, whether suicidal behavior indicates the survey done by closed-ended "yes/no/maybe" response or refers to another assessment (e.g., the Psychiatric Interview).“- It is specified now that the suicidality was assessed through the Psychiatric Interview.

  1. „Line 155-156, it would be more appropriate to put the sentence in possible and not certain terms (For example, one could change "it means" to "it suggests").“- It is corrected accordingly.

Discussion

  1. „Line 237: a bibliographic reference is missing to support the sentence "The feeling of fear of COVID-19 is an inevitable companion of stress."“- The bibliographic references are inserted and the quoted sentence is modified accordingly.

  1. „Line 255: it would be appropriate to report the data for clarity in the text.“- I reported the actual results in the text as suggested.

  1. „Line 262: it would be appropriate to include a reference regarding the increased suicide rate in Croatia during the early period of the pandemic.“- The proper reference is included.

  1. „It would be appropriate to include a literature reference inherent in the link between socioeconomic status, being a resident of developing countries, and mental well-being.“- The appropriate reference is included.

  1. „Line 274: there is a typo, one point too many after the date '2021'.“- It has been corrected.

  1. „Line 283: there is a missing bibliographic reference to support the sentence.“- The proper reference is included.

  1. „Line 289: there is a typo in the word 'menatlly' (mentally).”- It has been corrected.

  1. „Line 338-339: cite studies to support the sentence "the feeling of fear, which, as is well known, has - a particularly pronounced social component, is contagious."- Proper study has been cited.

Text proofreading and styling was done within the scope of your publishing company's offer. I hope that now the text is more readable, understandable and stylishly written in native English.

Once again, I would like to say that I am grateful for the constructive comments. The quality of work has been raised to a higher level, indeed.

I hope that you too will be satisfied with the corrections in the manuscript according to your instructions. Also, I hope that a positive evaluation of the manuscript and acceptance for publication will follow.

Should you have any other comments, I would appreciate them.

Respectfully,

Dušan Kuljančić

07 May 2023

Round 2

Reviewer 1 Report

Comments and Suggestions for Authors

I went through the edited version and found a significant improvement in the manuscript. I don’t have any additional comments to make.

Comments on the Quality of English Language

none

Reviewer 2 Report

Comments and Suggestions for Authors

Dear Author, congratulations on this work

Comments on the Quality of English Language

Dear Editors The article can be published